

# Large-eddy Simulation of a Wind-turbine Array subjected to Active Yaw Control

Mou Lin[1] and Fernando Porté-Agel[1]

[1]Wind Engineering and Renewable Energy Laboratory (WIRE), École Polytechnique Fédérale de Lausanne (EPFL), EPFL-ENAC-IIE-WIRE, CH-1015 Lausanne, Switzerland.

**Correspondence:** Fernando Porté-Agel (fernando.porte-agel@epfl.ch)

**Abstract.** This study validates the large-eddy simulation (LES) technique for the prediction of the flow through a wind turbine array subjected to active yaw control. The wind turbine array consists of three miniature wind turbines operated in both non-yawed and yawed configurations under full-wake and partial-wake conditions, for which wind tunnel flow measurements are available. The turbine-induced forces are parametrised by three different models: the standard actuator disk model (ADM-std), the blade element actuator disk model (ADM-BE), also referred to as the rotational actuator disk model (ADM-R), and the actuator line model (ALM). The time-averaged turbine power outputs and the profiles of the wake flow statistics (normalised streamwise mean velocity and streamwise turbulence intensity) obtained from the simulations using the ADM-std, the ADM-BE and the ALM are compared with experimental results. We find that simulations using the ADM-BE and ALM yield flow statistics that are in good agreement with the wind-tunnel measurements for all the studied configurations. In contrast, the results from LES with the ADM-std show discrepancies with the measurements under yawed and/or partial-wake conditions. These errors are due to the fact that the ADM-std assumes a uniform thrust force, thus failing to capture the inherently non-uniform distribution of the turbine-induced forces under partial wake conditions. In terms of power prediction, we find that LES using the ADM-BE yields better power prediction than the ADM-std and the ALM in both non-yawed and yawed conditions. As a result, we conclude that LES using the ADM-BE provides a good balance of accuracy and computational cost for simulations of the flow through wind farms subjected to AYC.

## 1 Introduction

As an indispensable part of the global transition to carbon neutrality, wind power has experienced a rapid growth worldwide in the past decades (GWEC, 2021). The majority of the wind power projects are developed in the form of wind farms, i.e., a cluster of wind turbines installed within a designated area, outputting the generated electricity to centralised substations before transmitting it to the grid. In comparison with distributed wind power, which consists of the installation of stand-alone turbines in different locations, developing wind energy in wind farms has many advantages, such as reducing the construction and maintenance overhead per turbine. On the other hand, wind turbines in wind farms often encounter the phenomenon of wake interference, i.e., wind turbines are exposed to the wakes of upwind turbines. This phenomenon can cause significant power losses and increase fatigue loads, and it has become the subject of many studies of wind farm flows (Barthelmie and



Jensen, 2010; Archer et al., 2018; Porté-Agel et al., 2020). Active yaw control (AYC), or active wake steering, is a novel wake-interference mitigation strategy that is drawing increasing interest in the research community. In this strategy, the upwind wind turbines are intentionally yawed to deflect their wakes away from downwind turbines. With a proper yawed configuration, the reduced power outputs in the yawed upwind turbines can be compensated by the increased power output in the downwind turbines. Therefore, a net power gain in the entire wind farm can be achieved.

Various early studies (Grant et al., 1997; Grant and Parkin, 2000) have revealed that the characteristics of the wake of a yawed turbine are significantly different from its non-yawed counterpart. Most notably, the yawed wake is deflected to the downwind-inclined side of the rotor. Dahlberg (2005) first indicated the potential of exploiting this phenomenon to optimise wind farm power using active yaw control, and he validated this concept with wind tunnel experiments. Since then, there has been a push in the wind energy community towards understanding the wake characteristics of yawed turbines. Jiménez et al. (2010) first derived an analytical wake model based on the top-hat velocity profile as an extension to the well-known Jensen wake model (Jensen, 1983) for non-yawed turbines. Bastankhah and Porté-Agel (2016) performed a wind tunnel study of a yawed miniature wind turbine in a turbulent boundary layer flow, and they found that the time-averaged profiles of the velocity deficit and the wake skew angle are Gaussian and self-similar in the far wake region. Exploiting this phenomenon, they developed a closed-form analytical model for the velocity deficit profiles of yawed turbines. Comparing with the top-hat Jimenez model, they found that the Gaussian model results are in better agreement with the measurements. Zong and Porté-Agel (2020a) developed a momentum-conserving method to superpose the wake velocity deficits behind multiple yawed turbines. Qian and Ishihara (2018) developed a bi-Gaussian parametric model for the turbulence intensity distribution in the wake of a yawed turbine. In a follow-up study, Qian and Ishihara (2021) also proposed a superposition model for predicting turbulence intensity in the wakes of multiple yawed turbines. The Qian and Ishihara model is based on the principle of the linear sum of squares of the added turbulence intensity, and it introduces a parametric correction for partial-wake scenarios.

Another distinctive feature of the wake of a yawed turbine is the formation of a counter-rotating vortex pair (CVP), which is induced by the lateral forces applied by the yawed turbine. Howland et al. (2016) carried out wind tunnel experiments on a yawed permeable disk in laminar inflows. They found that the permeable disk's wake is significantly asymmetrical, or "curled", in the spanwise direction. The curled wake is deformed by the presence of the CVP. Bastankhah and Porté-Agel (2016) also observed the CVP in the wind tunnel study of a yawed miniature wind turbine immersed in a turbulent boundary-layer flow. The curled wake pattern can sustain itself beyond the near-wake region and can still be observed at the location where a downwind turbine can be installed. Motivated by these experimental results, researchers made several efforts to incorporate the physics of the CVP in yawed wake modelling. Shapiro et al. (2018) treated the yawed turbine as a surface with an elliptic vorticity distribution and used lifting line theory to model the CVP formation. Based on the vorticity distribution proposed by Shapiro et al. (2018), Martínez-Tossas et al. (2019) developed a curled-wake model by solving the linearised Euler equations. King et al. (2021) derived an analytical approximation of the model of Martínez-Tossas et al. (2019) and formulated a reduced-order curled wake model that is computationally efficient. Zong and Porté-Agel (2020b) investigated the physics of the CVP in wind tunnel experiments and developed a point-vortex transportation model that reproduces the formation mechanism of the top-down asymmetric kidney-shaped wake behind a yawed turbine.





Besides experimental and theoretical approaches, numerical modelling is also a popular approach among researchers study-
ing AYC. Large-eddy simulation (LES), due to its relatively high fidelity, is widely used to investigate wind turbine wakes. In
LES, the turbine-induced forces can be represented by three main types of models. Jiménez et al. (2010) first used a standard
actuator disk model (ADM-std), which assumes a uniform distribution of the thrust force on the rotor disk, to parametrise the
yawed turbine-induced forces in LES. The ADM-std was also adopted by other researchers studying the wakes of multiple
turbines (Munters and Meyers, 2018; Stevens et al., 2018; Boersma et al., 2019). As an improvment to the ADM-std, the
blade element actuator disk model (ADM-BE), also referred to as the rotational actuator disk model (ADM-R), is proposed by
Wu and Porté-Agel (2011) and Porté-Agel et al. (2011), which uses the blade element theory to parametrise the non-uniform
thrust and tangential forces on the turbine rotor in LES. The ADM-BE was later applied by Fleming et al. (2018) to study the
large-scale trailing vortices in yawed wind turbine wakes. The actuator line model (ALM) is also a widely used method in LES
studies of yawed turbines (Fleming et al., 2016; Wang et al., 2017; Stevens et al., 2018; Archer and Vasel-Be-Hagh, 2019).
The ALM parametrises the rotor-induced forces on line elements distributed along each blade. Unlike LES using the ADM,
LES using the ALM can produce the tip vortices in the near wake region. However, LES using the ALM also requires higher
temporal and spatial resolution than the ADM counterpart (Martínez-Tossas et al., 2017), thus consuming substantially more
computational resources.

Lin and Porté-Agel (2019) have previously validated an LES framework using the ADM-BE to simulate the wake of a stand-
alone wind turbine subjected to AYC. Since the ultimate goal of AYC is to be applied to wind farms, it is natural to extend
the validation to multiple turbines. This study compares the results of LES using different turbine parametrisations (ADM-std,
ADM-BE and ALM) with wind tunnel measurements of a three-turbine array (Zong and Porté-Agel, 2021) in different turbine
layouts and yawed configurations.

The rest of the paper is structured as follows: Section 2 discusses the numerical configurations used in the simulations and
the methodology for evaluating the power output. Section 3 presents the simulation results obtained from LES using different
turbine parametrisations and compares them with wind tunnel measurements. Section 4 presents the conclusions drawn from
these results and discusses the possible extension of this work.

## 2  Methodology

### 2.1  Governing equations

A GPU-accelerated version of the WiRE-LES code is used in this study. The code has been developed at the Wind Engineering
and Renewable Energy Laboratory (WiRE) of the École Polytechnique Fédérale de Lausanne (EPFL), and it has been used and
validated in previous studies of wind turbine wakes, e.g., in Wu and Porté-Agel (2011), Porté-Agel et al. (2011), Abkar and
Porté-Agel (2015) and Lin and Porté-Agel (2019).



The WiRE-LES solves the spatially filtered incompressible Navier-Stokes (N-S) equations:

$$\frac{\partial \widetilde{u}_i}{\partial x_i} = 0,$$

$$\frac{\partial \widetilde{u}_i}{\partial t} + \widetilde{u}_j \left( \frac{\partial \widetilde{u}_i}{\partial x_j} - \frac{\partial \widetilde{u}_j}{\partial x_i} \right) = -\frac{\partial \widetilde{p}^*}{\partial x_i} - \frac{\partial \tau_{ij}}{\partial x_j} - \frac{f_i}{\rho} + \frac{F_p}{\rho} \delta_{i1}, \tag{1}$$

in which $\widetilde{u}_i$ is the spatially filtered velocity ($i = 1, 2, 3$ representing the streamwise, spanwise and vertical directions, respectively); $\widetilde{p}^*$ is the modified kinematic pressure; $f_i$ is the body force exerted by the wind turbine on the flow; $F_p$ is the pressure gradient imposed to drive the flow; $\tau_{ij} = \widetilde{u_i u_j} - \widetilde{u}_i \widetilde{u}_j$ is the kinematic sub-grid scale (SGS) stress, and it is parametrised using the modulated gradient model (MGM) proposed by Lu and Porté-Agel (2010):

$$\tau_{ij} = 2k_{sgs} \left( \frac{\tilde{G}_{ij}}{\tilde{G}_{kk}} \right), \tag{2}$$

in which $\tilde{G}_{ij}$ is defined as follow:

$$\tilde{G}_{ij} = \frac{\tilde{\Delta}_x^2}{12} \frac{\partial \tilde{u}_i}{\partial x} \frac{\partial \tilde{u}_j}{\partial x} + \frac{\tilde{\Delta}_y^2}{12} \frac{\partial \tilde{u}_i}{\partial y} \frac{\partial \tilde{u}_j}{\partial y} + \frac{\tilde{\Delta}_z^2}{12} \frac{\partial \tilde{u}_i}{\partial z} \frac{\partial \tilde{u}_j}{\partial z}, \tag{3}$$

$k_{sgs}$ is the zero-clipped SGS kinetic energy:

$$k_{sgs} = \mathbf{1}_{\tilde{G}_{ij}\tilde{S}_{ij}<0}(\tilde{G}_{ij}\tilde{S}_{ij}) \frac{4\tilde{\Delta}^2}{C_\epsilon^2} \left( -\frac{\tilde{G}_{ij}}{\tilde{G}_{kk}} \tilde{S}_{ij} \right)^2, \tag{4}$$

in which $\mathbf{1}_{\tilde{G}_{ij}\tilde{S}_{ij}<0}(\tilde{G}_{ij}\tilde{S}_{ij})$ is an indicator function taking the value of 1 if $\tilde{G}_{ij}\tilde{S}_{ij} < 0$ and 0 if $\tilde{G}_{ij}\tilde{S}_{ij} \geq 0$; $\tilde{S}_{ij}$ is the filtered strain rate; $\tilde{\Delta}$ is defined as $\sqrt[3]{\tilde{\Delta}_x \tilde{\Delta}_y \tilde{\Delta}_z}$, in which $\tilde{\Delta}_x$, $\tilde{\Delta}_y$ and $\tilde{\Delta}_z$ are the filter widths in the streamwise, spanwise and vertical directions. $C_\epsilon = 1.6$ is the model coefficient obtained from the simulations of the ABL flow using dynamic procedures (Lu and Porté-Agel, 2014).

## 2.2 Wind turbine parametrisation

In the WiRE-LES, three different types of wind turbine parametrisation are implemented (Fig. 1): the ADM-std, the ADM-BE and the ALM. In the ADM-std, a wind turbine is modelled as a permeable disk with thrust forces uniformly distributed within the rotor diameter. The magnitude of the thrust force is computed as:

$$F_x = \frac{1}{2} \rho A C_T U_{in}^2, \tag{5}$$

in which $\rho$ is the air density; $A$ is the sweeping area of the rotor disk; $C_T$ is the thrust coefficient of the wind turbine, and $U_{in}$ is the incoming wind speed. Since the turbines in wind farms often operate in the wakes of upwind turbines, their incoming velocities are retrieved as follow:

$$U_{in} = (1 - a) U_{loc}, \tag{6}$$





in which $U_{loc}$ is the local disk-averaged velocity at the rotor, and $a$ is the induction factor estimated from the thrust coefficient:

115

$$a = \frac{1}{2}(1 - \sqrt{1 - C_T}). \tag{7}$$

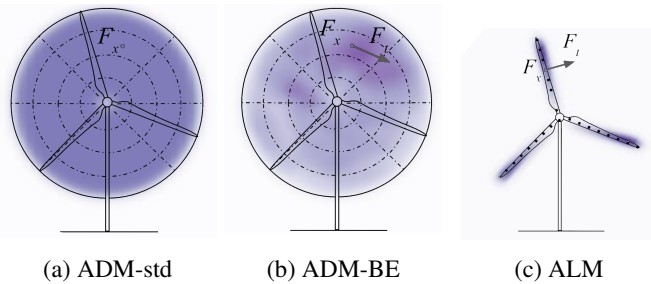

|            |            |          |
|:----------:|:----------:|:--------:|
| (a) ADM-std | (b) ADM-BE | (c) ALM |

**Figure 1.** Schematic representation of the three wind turbine parametrisations used in WiRE-LES. To illustrate the differences in the distribution of the forces computed using the three models, the normalised contours of the instantaneous force distribution (normalised by the respective maximum value) induced by each model are plotted.

In the ADM-BE, the turbine-induced forces are parametrised using the blade element theory. In contrast with the ADM-std, the forces in the ADM-BE are computed from the local velocity information and the aerodynamic properties of each blade element. As a result, the forces are non-uniform across the rotor. Furthermore, the ADM-BE not only takes the thrust forces into account but also models the tangential forces on the rotor. As a result, the ADM-BE introduces wake rotation in the wake of a turbine. After subdividing the rotor into an axisymmetric grid, the ADM-BE computes the local thrust force $F_x$ and the local tangential force $F_t$ as follow:

$$
\begin{aligned}
F_x &= \frac{1}{2}\rho U_{ref}^2 c\sigma \Phi(C_L \cos(\phi) + C_D \sin(\phi)), \\
F_t &= \frac{1}{2}\rho U_{ref}^2 c\sigma \Phi(C_D \cos(\phi) - C_L \sin(\phi)),
\end{aligned} \tag{8}
$$

in which $U_{ref}$ is the resultant inflow velocity at a given blade section; $c$ is the chord length, and $\sigma$ is the solidity of the blade section; $\Phi$ is the Prantl tip-loss correction factor; $\phi$ is the angle between the relative axial and the tangential velocity components at the blade element; $C_L$ and $C_D$ are the lift and drag coefficients interpolated from a 2D tabular dataset using the angle of attack (AoA) at a given blade element. A more detailed description of the ADM-BE and its application in yawed turbines can be found in Wu and Porté-Agel (2011) and Lin and Porté-Agel (2019).

The ALM computes the turbine-induced forces on line elements distributed on the moving turbine blades. The normal and the tangential forces on each source point are also computed from the blade element theory:

$$
\begin{aligned}
F_x &= \frac{1}{2}\rho U_{ref}^2 cw\Phi(C_L \cos(\phi) + C_D \sin(\phi)), \\
F_t &= \frac{1}{2}\rho U_{ref}^2 cw\Phi(C_D \cos(\phi) - C_L \sin(\phi)).
\end{aligned} \tag{9}
$$





Notice that the solidity $\sigma$ in the ADM-BE equations are replaced by the width of the blade sections $w$ in the ALM equations.

## 2.3 Case configuration

In this study, four simulation cases are set up to reproduce the boundary-layer wind tunnel experiments of a wind turbine array

subjected to active yaw control described by Zong and Porté-Agel (2021). The wind turbine array consists of three WiRE-01 miniature wind turbines. The diameter of the turbine $D = 0.15$ m, and the hub height $Z_{hub} = 0.125$ m. Each turbine is separated from the closest neighbouring turbines by a constant distance $S_x = 5D$ in the streamwise direction.

The configurations of the cases are summarized in Table 1. In Cases 1 and 2, the turbine rotor locations are aligned in the streamwise direction (i.e., lateral offset $S_y = 0D$), while a lateral offset $S_y = D/3$ is applied in Cases 3 and 4. In Cases 1 and

140 3, no active yaw control is applied (i.e., zero yaw angle for all turbines), while yawing configurations of $(25°, 15°, 0°)$ and $(20°, 20°, 0°)$ are applied in Cases 2 and 4, respectively. These were found to be the optimal yawing strategies that maximised the overall power output from the experiments (Zong and Porté-Agel, 2021). The wind turbine rotational speeds $\omega$ are also chosen to match those of the experiments.

**Table 1.** Case configurations of the wind tunnel experiments, with the specifications of the lateral offset $S_y$, the yaw angles $\gamma = (\gamma_1, \gamma_2, \gamma_3)$ and the rotational speeds $\omega = (\omega_1, \omega_2, \omega_3)$.

| No. | $S_y$ | $\gamma$ | $\omega$ (RPM) |
|---|---|---|---|
| Case 1 | $0D$ | $(0°, 0°, 0°)$ | (2183, 1405, 1560) |
| Case 2 | $0D$ | $(25°, 15°, 0°)$ | (2113, 1666, 1744) |
| Case 3 | $D/3$ | $(0°, 0°, 0°)$ | (2156, 1639, 1755) |
| Case 4 | $D/3$ | $(20°, 20°, 0°)$ | (2094, 1824, 2072) |

Schematics of the simulation domain are shown in Fig. 2. The size of the domain in the streamwise direction is $21.3D$. To

145 minimise the blockage effect, the size of the simulation domain is $10.7D$ in the spanwise direction and $5.3D$ in the vertical direction. The pressure gradient is imposed up to the height $Z_{bl} = 0.3$ m to create a boundary layer with the same height as in the experiments. The friction velocity $u_* = 0.26$ m s$^{-1}$ and the roughness length $z_0 = 10^{-4}$ m in the LES cases are chosen so that the streamwise mean inflow velocity and the streamwise turbulence intensity at the hub-height match the wind tunnel measurements (Fig. 3).

## 2.4 Numerical configuration

In the WiRE-LES, the spatially filtered N-S equations are solved by the pseudospectral method in the horizontal directions and by the second-order finite-difference method in the vertical direction. Explicit time integration is carried out using the Adams–Bashforth method. Such a choice of numerical schemes has also been applied and validated in previous wind turbine wake flow studies (Wu and Porté-Agel, 2011; Lin and Porté-Agel, 2019).



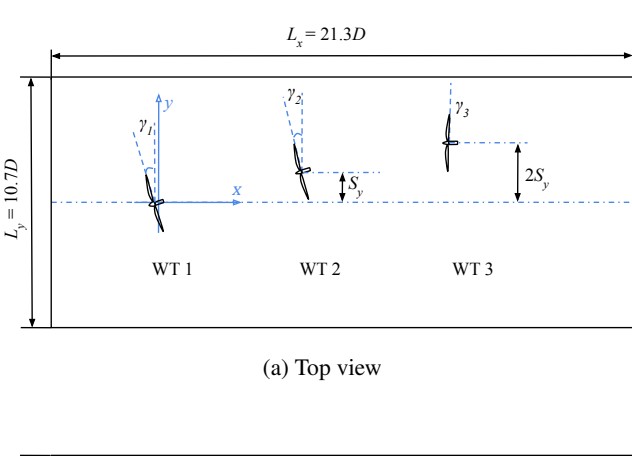

(a) Top view

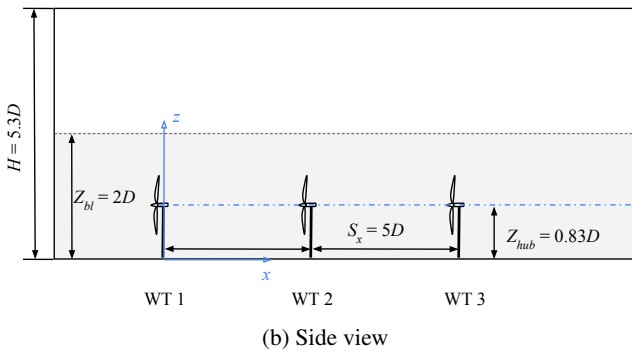

(b) Side view

**Figure 2.** Schematic plots of the simulation domain (not to scale): (a) top view (b) side view.

The simulation domain is discretised into a uniform mesh with the cell numbers of $256 \times 128 \times 128$ in the streamwise, spanwise and vertical directions, respectively. Since the 3/2 rule is applied in the spectral filter in the horizontal directions for the de-aliasing, the ratio between the filter size $\tilde{\Delta}$ and the grid size ($\Delta$) in the horizontal directions is $\tilde{\Delta}_x/\Delta_x = \tilde{\Delta}_y/\Delta_y = 1.5$. In the vertical direction, the ratio is $\tilde{\Delta}_z/\Delta_z = 1$. Therefore, the aspect ratio of the grid is $\Delta_x : \Delta_y : \Delta_z = 2 : 2 : 1$ and the aspect ratio of the filter is $\tilde{\Delta}_x : \tilde{\Delta}_y : \tilde{\Delta}_z = 3 : 3 : 1$. The ratios between the rotor diameter and the filter widths are $D/\tilde{\Delta}_y = 8$

and $D/\tilde{\Delta}_z = 24$ in the spanwise direction $y$ and the vertical direction $z$, respectively. The time step is chosen such that the Courant number is kept around 0.1.

     Periodic boundary conditions are used on the lateral boundaries in the horizontal directions ($x$ and $y$). On the vertical direction ($z$), a slip-wall condition is imposed on the top boundary and a non-penetration wall is applied on the bottom boundary with specified stress based on the logarithmic law of the wall. A precursor method is used to generate the turbulent inflow for

the simulation (Wu and Porté-Agel, 2011; Porté-Agel et al., 2013; Abkar and Porté-Agel, 2015), and a shifting boundary method is applied (Munters et al., 2016) at the inflow to mitigate the the formation of spurious locked-in streak-like structure (Fang and Porté-Agel, 2015).



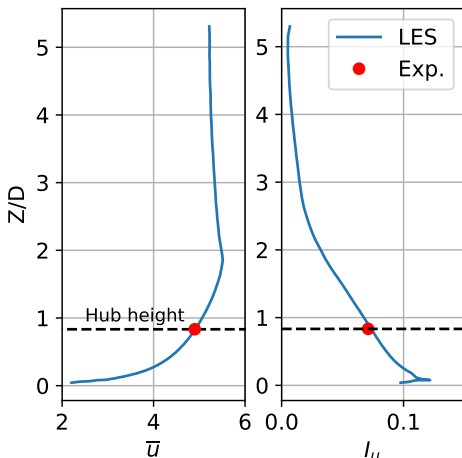

**Figure 3.** Vertical profiles of the streamwise mean velocity $\overline{u}$ and the streamwise turbulence intensity $I_u$. Blue solid lines represent the LES results and red dots represent the corresponding measurement data at the hub-height.

## 3 Results

### 3.1 Mean velocity

For the cases with zero lateral offset (Cases 1 and 2), contours of the normalised streamwise mean velocity in the $x-y$ plane at hub height are shown in Fig. 4. In Case 1, the turbines are not yawed, and the turbine array is aligned with the inflow direction. The second and the third turbines are fully exposed to the wakes of their upwind turbines. In Case 2, with the yaw angles $\gamma = (25°, 15°, 0°)$, the wakes of the yawed turbines are redirected to the side where the turbine rotor plane is inclined into the downwind direction. As a result, the second and the third turbines in Case 2 are partially exposed to the wake of their respective
upwind turbines.

Spanwise profiles of the normalised streamwise mean velocity at hub height are shown in Fig. 5. Behind the first turbine, we find that the maximum velocity deficits are slightly underestimated by LES using the ADM-std in the near wake for both non-yawed (Fig. 5a) and yawed configurations (Fig. 5b). As the wake develops further downstream, the results of the three models converge to the measurements. Behind the second turbine, the wakes of the turbine parametrised by the ADM-std have
180 slightly larger velocity deficits and wake widths compared to the measurements in the non-yawed configuration (Fig. 5c). In the yawed configuration (Fig. 5d), the velocity deficits in the ADM-std results are overestimated on the side where the turbine rotor is inclined downwind and are underestimated on the upwind-inclined side. As a result, the velocity profiles are further shifted to the negative spanwise ($y$) direction compared to the measurements. Behind the third turbine, the three models yield reasonable predictions of the mean velocity in the non-yawed configuration (Fig. 5e), while the ADM-std produces again an
185 unrealistic shift in the velocity profiles in the yawed configuration (Fig. 5f).



**Figure 4.** Contours of the normalised streamwise mean velocity $\overline{u}/\overline{u}_{hub}$ in the $x-y$ plane at hub height obtained from the wind-tunnel experiments and LES using the ADM-std, ADM-BE and ALM. The lateral offset of the turbines is zero (Cases 1 and 2).

Fig. 6 and Fig. 7 show a comparison between measured and simulated contours and spanwise profiles of the mean velocity, respectively, for the partial-wake cases under consideration. Due to the lateral offset of the turbines, the second and the third turbines are partially exposed to the incoming wakes in both non-yawed and yawed configurations. In Cases 3 and 4, where the partial-wake condition occurs, shifted velocity profiles with respect to the measurements are observed in the wakes of the second and the third turbines in the ADM-std results. Furthermore, an underestimation of the velocity deficits is also observed in the wake of the third turbine in the ADM-std results of Case 4.





(a) Wind turbine 1, Case 1 $(0°, 0°, 0°)$      (b) Wind turbine 1, Case 2 $(25°, 15°, 0°)$

(c) Wind turbine 2, Case 1 $(0°, 0°, 0°)$      (d) Wind turbine 2, Case 2 $(25°, 15°, 0°)$

(e) Wind turbine 3, Case 1 $(0°, 0°, 0°)$      (f) Wind turbine 3, Case 2 $(25°, 15°, 0°)$

**Figure 5.** Spanwise profiles of the normalised streamwise mean velocity $\overline{u}/\overline{u}_{hub}$ in the $x-y$ plane at hub height obtained from the wind-tunnel experiments, LES using the ADM-std, ADM-BE and ALM. The lateral offset of the turbines is zero (Cases 1 and 2).

Fig. 8 shows the trajectories of the location of the maximum velocity deficit in the wake, in different configurations. The trajectories of the ADM-std results are shifted from the measurements behind the turbines in the partial wake condition. This is consistent with the shifted pattern of the ADM-std results observed in the velocity profile plots (Fig. 5 and Fig. 7). This issue can

be explained by a key model assumption of the ADM-std: the turbine-induced forces are modelled as thrust forces uniformly distributed on the rotor disk. When a wind turbine is partially exposed to the incoming wake, the ADM-std overestimates the thrust force in the rotor sections with more wake exposure and underestimates the thrust forces in the less-exposed sections. As a result, the velocity deficits are overestimated behind the rotor sections with more exposure to the incoming wake and underestimate behind the less-exposed section. This, in turn, leads to the observed spurious lateral shift in the location of the





maximum velocity deficit. On the other hand, the ADM-BE and the ALM both resolve the non-homogeneous force distribution on the rotor, producing results that are in better agreement with the measurements in the partial-wake condition, characterized by the inhomogeneous inflows to the turbine.



(a) Experiment, Case 3 $(0°, 0°, 0°)$

(b) Experiment, Case 4 $(20°, 20°, 0°)$

(c) ADM-std, Case 3 $(0°, 0°, 0°)$

(d) ADM-std, Case 4 $(20°, 20°, 0°)$

(e) ADM-BE, Case 3 $(0°, 0°, 0°)$

(f) ADM-BE, Case 4 $(20°, 20°, 0°)$

(g) ALM, Case 3 $(0°, 0°, 0°)$

(h) ALM, Case 4 $(20°, 20°, 0°)$

**Figure 6.** Top-view contours of the normalised streamwise mean velocity $\overline{u}/\overline{u}_{hub}$ in the $x-y$ plane at hub height obtained from the wind-tunnel experiments, LES using the ADM-std, ADM-BE and ALM. The wind turbines are offset in the spanwise direction with a distance of $D/3$ (Cases 3 and 4).

.







(a) Wind turbine 1, Case 3 $(0°, 0°, 0°)$

(b) Wind turbine 1, Case 4 $(20°, 20°, 0°)$

(c) Wind turbine 2, Case 3 $(0°, 0°, 0°)$

(d) Wind turbine 2, Case 4 $(20°, 20°, 0°)$

(e) Wind turbine 3, Case 3 $(0°, 0°, 0°)$

(f) Wind turbine 3, Case 4 $(20°, 20°, 0°)$

**Figure 7.** Profiles of the normalised streamwise mean velocity $\overline{u}/\overline{u}_{hub}$ in the $x - y$ plane at hub height obtained from the wind-tunnel experiments, LES using the ADM-std, ADM-BE and ALM. The wind turbines are offset in the spanwise direction with a distance of $D/3$ (Cases 3 and 4).

### 3.2 Turbulence intensity

Contours and profiles of the streamwise turbulence intensity in the $x - y$ plane at hub height are shown in Fig. 9 and Fig. 10, respectively. The experiment results of the non-yawed and yawed configurations are compared with the corresponding LES results using the ADM-std, ADM-BE and the ALM. Since the wind tunnel measurements of the turbulence intensity are not available for Case 3 and Case 4, we only analyse Case 1 and Case 2 with zero offset.

In the measurement contours shown in Figures 9a and 9b, large magnitude of turbulence intensity is observed at the edges of the wake due to the strong shear in these regions. In the non-yawed case (Figures 9a), the turbulence intensity in the wakes





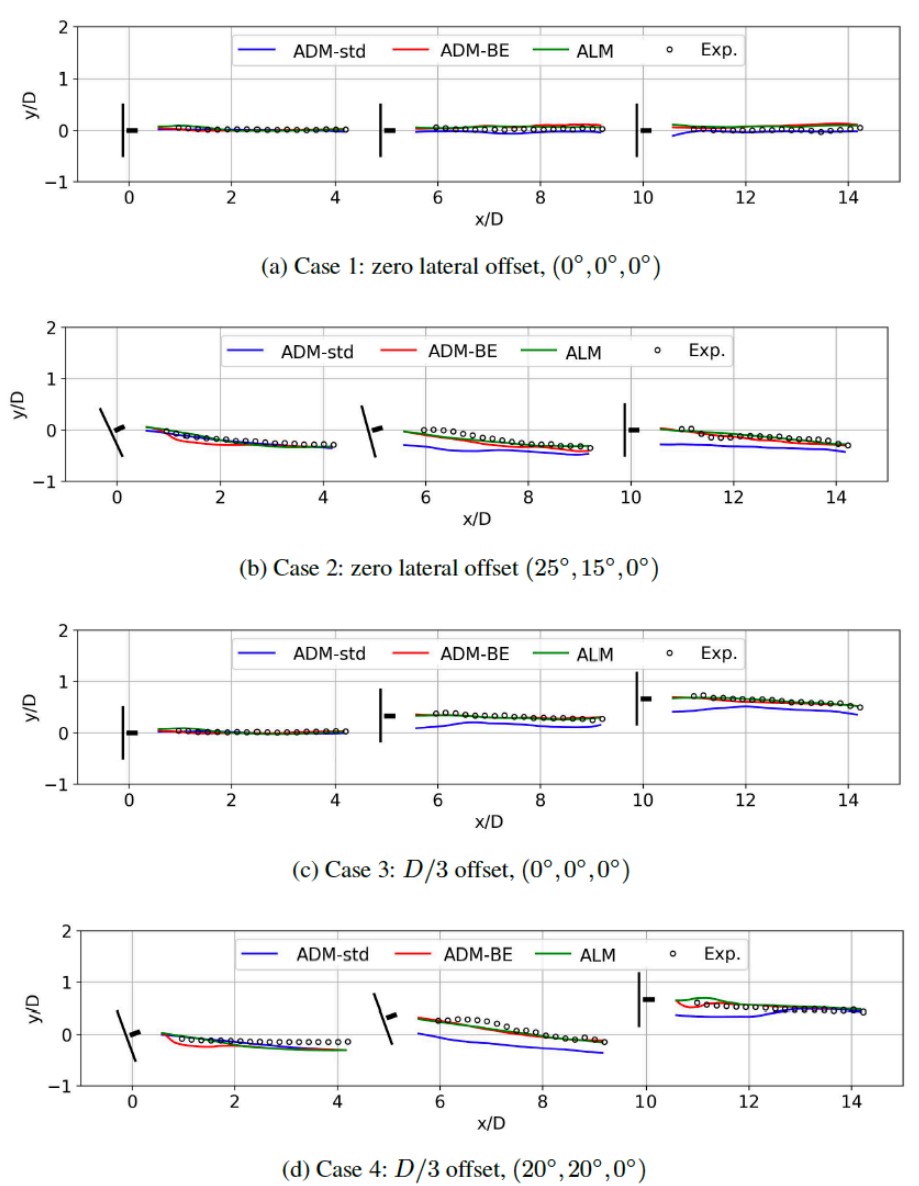

**Figure 8.** Trajectories of maximum velocity deficit location obtained from the wind-tunnel experiments, LES using the ADM-std, ADM-BE and ALM.

is largely symmetric with respect to the wake center-line. In the yawed case (Fig. 9b), the turbulence intensity on the positive $y$ side of the wake is larger than the turbulence intensity on the negative $y$ side.

By comparing the LES results with the measurements in the turbulence intensity contours (Fig. 9), we find that the results of LES using the ADM-std show discrepancies with the measurements in the yawed case with the partial-wake condition. In the





(a) Experiment, Case 1 $(0°, 0°, 0°)$         (b) Experiment, Case 2 $(25°, 15°, 0°)$

(c) ADM-std, Case 1 $(0°, 0°, 0°)$         (d) ADM-std, Case 2 $(25°, 15°, 0°)$

(e) ADM-BE, Case 1 $(0°, 0°, 0°)$         (f) ADM-BE, Case 2 $(25°, 15°, 0°)$

(g) ALM, Case 1 $(0°, 0°, 0°)$         (h) ALM, Case 2 $(25°, 15°, 0°)$

**Figure 9.** Top-view contours of the turbulence intensity $I_u$ in the $x - y$ plane at hub height obtained from the wind-tunnel experiments, LES using the ADM-std, ADM-BE and ALM. The lateral offset of the turbines is zero (Cases 1 and 2).

wakes behind the second and the third turbine, LES using the ADM-std overestimates the turbulence intensity with respect to
the measurements on the negative $y$ side of the wake. This is consistent with the overestimation of the mean velocity gradient
in the ADM-std results on the positive $y$ side of the skewed wake (Fig. 5). Furthermore, in comparison with LES using the
ADM-BE and the ALM, LES using the ADM-std underestimates the magnitude of the turbulent flux $\overline{u'v'}$ on the positive $y$ side
of the wake. Since the turbulence production term is defined by taking a product of the velocity gradient and the turbulence
flux, such differences in the results of LES using the ADM-std lead to an incorrect turbulence intensity distribution in the
partial-wake scenario. Comparisons of the turbulence intensity profiles in Fig. 10 also show that LES using the ADM-std, the
ADM-BE and the ALM slightly overspread the turbulence in the wakes: the turbulence intensity profiles of the LES results





are wider than the measurements in both non-yawed and yawed cases. This phenomenon is caused by the fact that the turbine forces in the LES are smeared by smoothing kernels in the turbine parametrisations. As a result, the shear layer produced at the edges of the wakes is wider compared to the the measurements, causing the wider turbulence intensity profiles in the LES

results.

(a) Wind turbine 1, Case 1 $(0°, 0°, 0°)$

(b) Wind turbine 1, Case 2 $(25°, 15°, 0°)$

(c) Wind turbine 2, Case 1 $(0°, 0°, 0°)$

(d) Wind turbine 2, Case 2 $(25°, 15°, 0°)$

(e) Wind turbine 3, Case 1 $(0°, 0°, 0°)$

(f) Wind turbine 3, Case 2 $(25°, 15°, 0°)$

**Figure 10.** Profiles of the streamwise turbulence intensity $I_u$ in the $x - y$ plane at hub height obtained from the wind-tunnel experiments, LES using the ADM-std, ADM-BE and ALM.




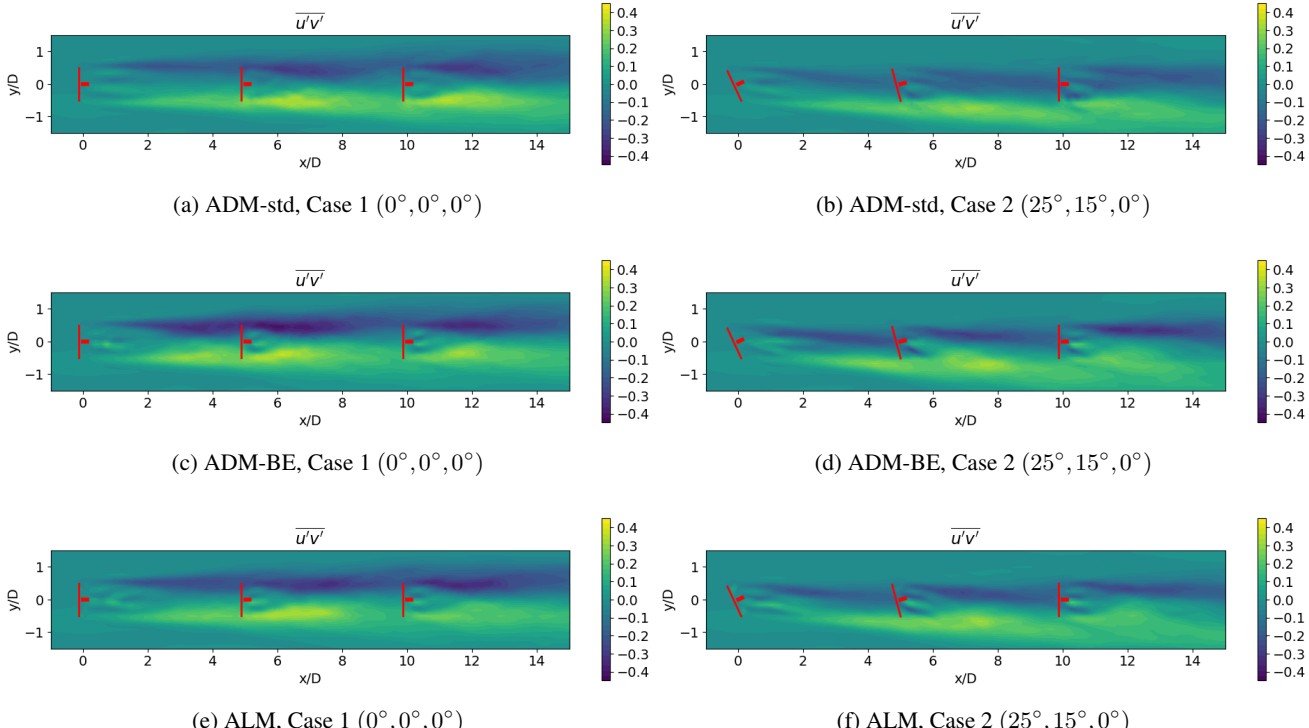

(a) ADM-std, Case 1 $(0°, 0°, 0°)$

(b) ADM-std, Case 2 $(25°, 15°, 0°)$

(c) ADM-BE, Case 1 $(0°, 0°, 0°)$

(d) ADM-BE, Case 2 $(25°, 15°, 0°)$

(e) ALM, Case 1 $(0°, 0°, 0°)$

(f) ALM, Case 2 $(25°, 15°, 0°)$

**Figure 11.** Top-view contours of the turbulent flux $\overline{u'v'}$ $(\mathrm{m}^2/\mathrm{s}^2)$ in the $x - y$ plane at hub height obtained from LES using the ADM-std, ADM-BE and ALM. The lateral offset of the turbines is zero (Cases 1 and 2).

### 3.3 Normalised power output

Finally, we compare the power outputs obtained from the ADM-std, the ADM-BE and the ALM with the power measured in the wind tunnel experiments. The normalised power outputs (normalised by the power output of the non-yawed turbine facing an undisturbed inflow) of the zero and the $D/3$ offset cases are shown in Fig. 12.

In Case 1, with zero lateral offset and zero yaw angle, the ADM-BE yields the smallest errors with respect to the measurements while the ADM-std yields the largest. In Case 2, with the yaw angles of $(25°, 15°, 0°)$, the ADM-std and the ADM-BE yield similar power output in the first and the third turbine, while the ADM-BE has the largest underestimation in the second turbine. The ALM, on the other hand, has the largest overestimation in the first turbine but the least in the third turbine. In Case 3 and Case 4, with a D/3 lateral offset, the ADM-BE outperform the ADM-std and the ALM and have the smallest errors in

the power outputs in both non-yawed and yawed configurations.

As shown in Fig. 12, the ADM-BE, which explicitly resolves the torque, and therefore the power, yields more accurate power predictions than than the ADM-std. The inaccuracies in the ADM-std results are largely due to the fact that the power output is calculated using the power curve. Since that curve is obtained from wind turbines operating in homogeneous inflow and









(a) Case 1: zero lateral offset, yaw angles $(0°, 0°, 0°)$

(b) Case 1: errors in the normalised power

(c) Case 2: zero lateral offset, yaw angles $(25°, 15°, 0°)$

(d) Case 2: errors in the normalised power

(e) Case 3: $D/3$ offset, yaw angles $(0°, 0°, 0°)$

(f) Case 3: errors in the normalised power

(g) Case 4: $D/3$ offset, yaw angles $(20°, 20°, 0°)$

(h) Case 4: errors in the normalised power

**Figure 12.** Normalised power outputs $\tilde{P}_i = P_i/P_1$: the power output of $i$th turbine $P_i$ is normalised by the power of the first turbine $P_1$. The error of the $\tilde{P}_i$ obtained from LES using the ADM-std, ADM-BE and ALM with respective to the experiments are shown in (b), (d), (f) and (h).




zero-yaw conditions, it is expected to be less accurate when the wind turbine operates in partial wake or yawed conditions. It
is also found that, in general, the ADM-BE outperforms the ALM, even if both of them are torque-resolving parametrisations.
This is consistent with previous studies (Martinez et al., 2012; Martínez-Tossas et al., 2017) that have shown that the power
prediction from the ALM is more sensitive to the smearing kernel used to project the localised blade-induced forces on the
Cartesian mesh grid, compared to the ADM. As a result, the ALM fails to yield good power predictions in the typical mesh
resolution ($\sim 10$ grid points along the rotor diameter) used in simulations of wind farm flows (Stevens et al., 2018).

## 245   4   Summary

In this study, we validate an LES framework with different wind turbine force parametrisations (ADM-std, ADM-BE and ALM)
for the prediction of the flow through a three-turbine array. The simulations are set to match existing wind tunnel experiments
for which flow and power measurements are available for different turbine lateral offsets (with respect to the wind direction)
and different active yaw control strategies.

Comparisons with wind tunnel measurements show that LES with wind turbine models that capture the local distribution
of the turbine-induced forces (ADM-BE and ALM) provide reasonably accurate predictions of the streamwise mean velocity
deficit and the streamwise turbulence intensity in the wakes of the three wind turbines for all the considered conditions of
lateral offset and yaw control. In contrast, the wake flows simulated with the standard actuator disk model (ADM-std) show
a lateral shift with respect to the measurements when the turbines are exposed to partial wake conditions produced by either
lateral offset of the turbines or/and active yaw control. This is due to the fact that the assumption of homogeneous thrust force
made by the ADM-std hinders the model from capturing the non-homogeneous force distribution experienced by the rotor and,
consequently, the correct wake velocity deficit distribution under partial wake conditions. Moreover, the standard ADM-std
yields the largest inaccuracies in the power output predictions due to the fact that it is based on the power curve, which is
not reliable under non-homogeneous inflow conditions (e.g., under partial wake conditions). The two torque-resolving models
(ADM-BE and ALM) are found to provide more accurate power predictions, with the ADM-BE showing less sensitivity to
errors associated with the smearing kernel used to project the forces on the grid.

From the aforementioned results, we conclude that the ADM-BE provides a good balance between accuracy and compu-
tational cost for the simulation of wind farm flows under different conditions, including partial wake and active yaw control.
In our future research, we plan to apply the validated LES framework to investigate optimal AYC strategies under different
atmospheric conditions, e.g., turbulence intensity and atmospheric stability. Furthermore, since the turbine forces are explicitly
resolved by the ADM-BE and ALM, the LES framework could also be applied to study structural loads in wind farms subjected
to AYC.

*Data availability.*   The data set is available on Zenodo (Lin and Porté-Agel, 2022)



*Author contributions.* ML contributed to the conceptualization, curation, formal analysis, investigation, methodology, software, visualization and writing (original draft). FPA contributed to the conceptualization, supervision, software, funding acquisition, project administration and writing (review & editing).

*Competing interests.* The authors declare that they have no conflict of interest.

*Acknowledgements.* This research was funded by the Swiss Federal Office of Energy (Grant SI/501337-01) and the Swiss National Science Foundation (Grant 200021_172538). Computing resources were provided by EPFL through the use of the facilities of Scientific IT and Application Support Center (SCITAS). The authors also would like to thank Dr Haohua Zong for providing helpful information on the wind tunnel measurements used in this study.





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
