# Peer review of "Large-eddy Simulation of a Wind-turbine Array subjected to Active Yaw Control"

_Wind Energy Science, 2022_

## Author Response (AR1)

**Response to Reviewer 1:**

We would like to thank the reviewer for his constructive advice. Our point-to-point responses to the reviewer's comments are listed below:

- *The images in the text are far (a few pages away) from where they are referenced. This might change with the final formatting of the manuscript, but the authors should try to adjust this whenever possible.*

  This is a pure format issue at this stage, as we are using the "manuscript" option in the LaTeX template provided by the journal. In the final submission, we will switch to the double-column option and the figures will be much closer to the text where the figures are referenced.

- *Comment: Please include the original reference for the ALM: https://asmedigitalcollection.asme.org/fluidsengineering/article/124/2/393/444521/Numerical-Modeling-of-Wind-Turbine-Wakes*

  We have added the reference (*Sorensen and Shen, 2002)* suggested by the reviewer and cited it in the introduction of the studies using the ALM (line 78, page 3 in the revised manuscript).

- *The excellent agreement between measurements and simulations is also influenced by the selection of grid resolution in the simulations. A few comments in the Summary highlighting the grid resolution and its effect would be useful.*

  To address the comment of the reviewer, we have added an appendix containing a mesh convergence analysis on the wake flow statistics obtained from the LES (line 270, page 19 in the revised manuscript).

- *Were you also able to obtain measurements as a function of height (z/D)? I would expect a similar level of agreement, but it would have been nice to see that.*

  The measurement data used in the study is extracted from *Zong and Porté-Agel (2021)*. Unfortunately, in that study the authors only performed PIV measurements on the horizontal plane at the hub height. Therefore, we are not able to compare the simulation results with the measurements as a function of height.

**Response to Reviewer 2:**

We would like to thank the reviewer for the constructive advice. Our point-to-point responses to the reviewer's comments are listed below:

— *"Line 25: active yaw control is referred to as a novel strategy. However, as is also clear from the introduction, this approach has been around for a while."*

The corresponding line (line 19, page 1 in the new manuscript) has been modified to *"Active yaw control (AYC), or active wake steering, is a wake-interference mitigation strategy that is drawing increasing interest in the research community."*

— *"The manuscript uses ADM-std, ADM-BE, and ALM as abbreviations for the used turbine models. These terms are different from previous works from the group, which is somewhat confusing. Also, figure 10 uses the notation used in previous publications by the group, so the notation is not consistent throughout the manuscript. In any case, the notation of the used turbine model should be made consistent in the manuscript itself."*

In this manuscript, we decided to update the terminologies from the ones used in previous studies to emphasise the main characteristics of the model. The typo in the legend of Fig. 10 (Fig. 11 in the new manuscript) pointed out by the reviewer has been fixed in the new manuscript.

— *"Can you please provide a description of, or reference to, the tabulated lift and drag coefficients and turbine thrust coefficients used in the simulations."*

The reference to the tabulated lift and drag coefficients and turbine thrust coefficients has been added (Revaz and Porté-Agel, 2021) in the new manuscript (line 9, page 4 in the new manuscript).

— *"line 159:160: the resolution in turbine diameters is mentioned in filter width grid spacing. It seems that this should just be the grid spacing."*

To avoid possible confusion, we have explicitly specified the two ratios in the new manuscript (line 71 - 79, page 4).

— *"The reference to the experimental data is incorrect. The Zong and Porte-Abel paper referenced their recent JFM and not the Renewable Energies paper that describes the corresponding experiments."*

We confirm that we are indeed using the experimental data from Zong and Porté-Agel (2021), published in Renewable Energy. In that study, the authors carried out wind tunnel measurements of multiple turbines in different configurations, corresponding to the simulations we set up in this study.

*— line 196-199: could you include a graph confirming the discussion started there?*

We have added a new figure (Fig. 8 in the new manuscript) containing the contours of the time-averaged thrust forces on a turbine partially exposed to the incoming wake (WT 3 in Case 2). The discussion here is also re-written to reflect the information shown in the new figure.

*— Figure 11 is not referenced in the text directly. It is shortly discussed around line 215 in the 'turbulence intensity section,' although this figure does not show 'turbulence intensity.'*

We have added a reference to this figure (Fig. 12 in the new manuscript) in the manuscript.

— line 240: the authors conclude ADM-BE is better than ALM. However, from figure 12, they seem to perform similarly, i.e. sometimes one, sometimes another model gives the best result.

— What are the exact data underlying the above statement, and is it significant with respect to uncertainties due to, for example, limited time averaging and the employed resolution?

— If necessary, update the abstract and conclusion with respect to the updated analysis on the ADM-BE and ALM comparison; see point above on line 240.

We would like to give a joint response to the three comments above. Firstly, we have fixed a sequence error in Fig. 12 of the old manuscript. The results of Case 4 were misplaced to the panels entitled with "Case 2", and vice versa. In the new manuscript, this misplacement has been fixed in Fig. 14 and Fig. 15.

To address the reviewer's comment on the uncertainty, we have added a new figure (Fig. 13 in the new manuscript) showing the power outputs and the relative errors of a zero-yaw stand-alone turbine from the simulations with different grid resolutions. For this case, the uncertainty level of the power measurement (4.5%) is available in the literature (Zong and Porté-Agel, 2021). We find that the advantage of the ADM-BE over the ADM-std and the ALM in the baseline grid is significant with respect to power measurement uncertainty. The ALM only produces a similar power output compared to the ADM-BE when the grid is refined. This observation supports our point that the ADM-BE is a model with a good balance of accuracy and computational cost.

As for the reviewer's comment on the time averaging, we have added a description of the averaging time (10 minutes) in line 76, page 4. We would like to point out that this averaging time window is more than three times longer than the one used in the experiment (3 minutes). Therefore, we conclude that the averaging time we used is long enough so that the uncertainty coming from the averaging has little impact on the result.

Finally, to address the reviewer's comments on the underlying data for comparing different models, we have updated the presentation of the power data in the new manuscript. In Fig. 14, the power

outputs are normalised by the measured power of a zero-yaw frontal turbine. The normalised power errors of each turbine are shown in Fig. 15, and the normalised total errors of the three-turbine array are shown in Fig. 16. The normalised total error is the metric we used to compare different models in each case. We hope the updated figures make it easier for readers to compare the power outputs of different models.

— "line 240 makes a comparison of ADM-BE and ALM. A statement follows that this is consistent with literature comparing ALM and ADM-std. This argumentation is not consistent. Please revise accordingly. "

We have revised the reference to Martínez-Tossas et al. (2015) from the conference paper (Martínez-Tossas et al., 2012) we cited previously and corrected this typo.